# 3D Scene Compression through Entropy Penalized Neural Representation Functions

## Abstract

Some forms of novel visual media enable the viewer to explore a 3D scene from essentially arbitrary viewpoints, by interpolating between a discrete set of original views. Compared to 2D imagery, these types of applications require much larger amounts of storage space, which we seek to reduce. Existing approaches for compressing 3D scenes are based on a separation of compression and rendering: each of the original views is compressed using traditional 2D image formats; the receiver decompresses the views and then performs the rendering. We unify these steps by directly compressing an implicit representation of the scene, a function that maps spatial coordinates to a radiance vector field, which can then be queried to render arbitrary viewpoints. The function is implemented as a neural network and jointly trained for reconstruction as well as compressibility, in an end-to-end manner, with the use of an entropy penalty on the parameters. Our method significantly outperforms a state-of-the-art conventional approach for scene compression, achieving simultaneously higher quality reconstructions and lower bitrates. Furthermore, we show that the performance at lower bitrates can be improved by jointly representing multiple scenes using a soft form of parameter sharing.

## 1 Introduction

The ability to render 3D scenes from arbitrary viewpoints can be seen as a big step in the evolution of digital multimedia, and has applications such as mixed reality media, graphic effects, design, and simulations. Often such renderings are based on a number of high resolution images taken of some original scene, and it is clear that to enable many applications, the data will need to be stored and transmitted efficiently over low-bandwidth channels (e.g., to a mobile phone for augmented reality).

Traditionally, the need to compress this data is viewed as a separate need from rendering. For example, light field images (LFI) consist of a set of images taken from multiple viewpoints. To compress the original views, often standard video compression methods such as HEVC (Sullivan et al., 2012) are repurposed (Jiang et al., 2017; Barina et al., 2019). Since the range of views is narrow, light field images can be effectively reconstructed by "blending" a smaller set of representative views (Astola & Tabus, 2018; Jiang et al., 2017; Zhao et al., 2018; Bakir et al., 2018; Jia et al., 2019). Blending based approaches, however, may not be suitable for the more general case of arbitrary-viewpoint 3D scenes, where a very diverse set of original views may increase the severity of occlusions, and thus would require storage of a prohibitively large number of views to be effective.

A promising avenue for representing more complete 3D scenes is through neural representation functions, which have shown a remarkable improvement in rendering quality (Mildenhall et al., 2020; Sitzmann et al., 2019; Liu et al., 2020; Schwarz et al., 2020). In such approaches, views from a scene are rendered by evaluating the representation function at sampled spatial coordinates and then applying a differentiable rendering process. Such methods are often referred to as implicit representations, since they do not specify explicitly the surface locations and properties within the scene, which would be required to apply some conventional rendering techniques like rasterization (Akenine-Möller et al., 2019). However, finding the representation function for a given scene requires training a neural network. This makes this class of methods difficult to use as a rendering method in the existing framework, since it is computationally infeasible on a low-powered end device like a mobile phone, which are often on the receiving side. Due to the data processing inequality, it may also be inefficient to compress the original views (the training data) rather than the trained

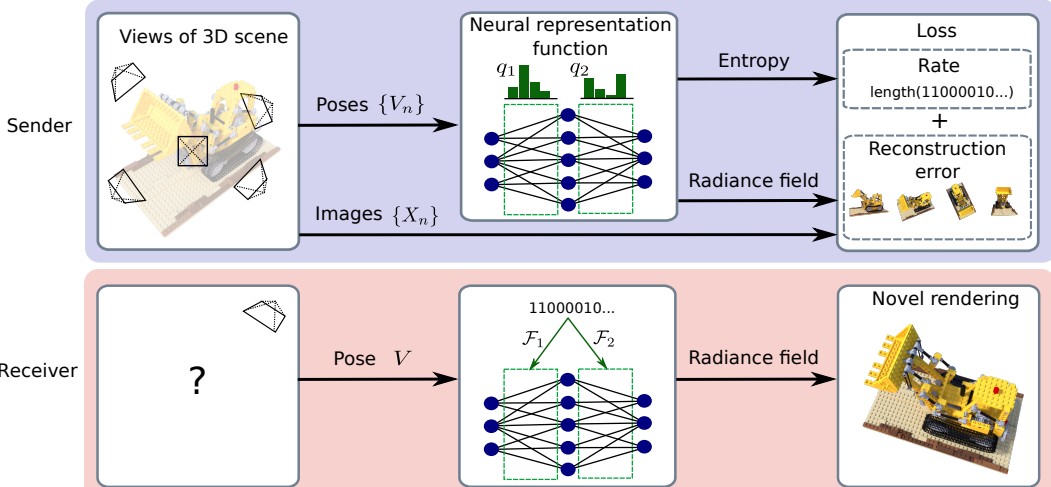

Figure 1: Overview of cNeRF. The sender trains an entropy penalized neural representation function on a set of views from a scene, minimizing a combination of rate and distortion. The receiver can use the compressed model to render novel views.

representation itself, because the training process may discard some information that is ultimately not necessary for rendering (such as redundancy in the original views, noise, etc.).

In this work, we propose to apply neural representation functions to the scene compression problem by compressing the representation function itself. We use the NeRF model (Mildenhall et al., 2020), a method which has demonstrated the ability to produce high-quality renders of novel views, as our representation function. To reduce redundancy of information in the model, we build upon the model compression approach of Oktay et al. (2020), applying an entropy penalty to the set of discrete reparameterized neural network weights. The *compressed NeRF* (cNeRF) describes a radiance field, which is used in conjunction with a differentiable neural renderer to obtain novel views (see Fig. 1). To verify the proposed method, we construct a strong baseline method based on the approaches seen in the field of light field image compression. cNeRF consistently outperforms the baseline method, producing simultaneously superior renders and lower bitrates. We further show that cNeRF can be improved in the low bitrate regime when compressing multiple scenes at once. To achieve this, we introduce a novel parameterization which shares parameters across models and optimize jointly across scenes.

## 2 BACKGROUND

We define a multi-view image dataset as a set of tuples $D = \{(V_n, X_n)\}_{n=1}^{N}$, where $V_n$ is the camera pose and $X_n$ is the corresponding image from this pose. We refer to the 3D ground truth that the views capture as the scene. In what follows, we first provide a brief review of the neural rendering and the model compression approaches that we build upon while introducing the necessary notation.

**Neural Radiance Fields (NeRF)**   The neural rendering approach of Mildenhall et al. (2020) uses a neural network to model a radiance field. The radiance field itself is a learned function $g_\theta : \mathbb{R}^5 \to (\mathbb{R}^3, \mathbb{R}^+)$, mapping a 3D spatial coordinate and a 2D viewing direction to a RGB value and a corresponding density element. To render a view, the RGB values are sampled along the relevant rays and accumulated according to their density elements. The learned radiance field mapping $g_\theta$ is parameterized with two multilayer perceptrons (MLPs), which Mildenhall et al. (2020) refer to as the "coarse" and "fine" networks, with parameters $\theta_c$ and $\theta_f$ respectively. The input locations to the coarse network are obtained by sampling regularly along the rays, whereas the input locations to the fine network are sampled conditioned on the radiance field of the coarse network. The networks are

trained by minimizing the distance from their renderings to the ground truth image:

$$L = \underbrace{\sum_{n=1}^{N} \left\| \hat{X}_n^c(\theta_c; V_n) - X_n \right\|_2^2}_{L_c(\theta_c)} + \underbrace{\sum_{n=1}^{N} \left\| \hat{X}_n^f(\theta_f; V_n, \theta_c) - X_n \right\|_2^2}_{L_f(\theta_f; \theta_c)} \tag{1}$$

Where $|| \cdot ||_2$ is the Euclidean norm and the $\hat{X}_n$ are the rendered views. Note that the rendered view from the fine network $\hat{X}_n^f$ relies on both the camera pose $V_n$ and the coarse network to determine the spatial locations to query the radiance field. We drop the explicit dependence of $L_f$ on $\theta_c$ in the rest of the paper to avoid cluttering the notation. During training, we render only a minibatch of pixels rather than the full image. We give a more detailed description of the NeRF model and the rendering process in Appendix Sec. A.

**Model Compression through Entropy Penalized Reparameterization** The model compression work of Oktay et al. (2020) reparameterizes the model weights $\Theta$ into a latent space as $\Phi$. The latent weights are decoded by a learned function $\mathcal{F}$, i.e. $\Theta = \mathcal{F}(\Phi)$. The latent weights $\Phi$ are modeled as samples from a learned prior $q$, such that they can be entropy coded according to this prior. To minimize the rate, i.e. length of the bit string resulting from entropy coding these latent weights, a differentiable approximation of the self-information $I(\phi) = -\log_2(q(\phi))$ of the latent weights is penalized. The continuous $\Phi$ are quantized before being applied in the model, with the straight-through estimator (Bengio et al., 2013) used to obtain surrogate gradients of the loss function. Following Ballé et al. (2017), uniform noise is added when learning the continuous prior $q(\phi + u)$ where $u_i \sim U(-\frac{1}{2}, \frac{1}{2}) \; \forall \; i$. This uniform noise is a stand-in for the quantization, and results in a good approximation for the self-information through the negative log-likelihood of the noised continuous latent weights. After training, the quantized weights $\tilde{\Phi}$ are obtained by rounding, $\tilde{\Phi} = \lfloor \Phi \rceil$, and transmitted along with discrete probability tables obtained by integrating the density over the quantization intervals. The continuous weights $\Phi$ and any parameters in $q$ itself can then be discarded.

## 3 METHOD

To achieve a compressed representation of a scene, we propose to compress the neural scene representation function itself. In this paper we use the NeRF model as our representation function. To compress the NeRF model, we build upon the model compression approach of Oktay et al. (2020) and jointly train for rendering as well as compression in an end-to-end trainable manner. We subsequently refer to this approach as cNeRF. The full objective that we seek to minimize is:

$$\mathcal{L}(\Phi, \Psi) = \underbrace{L_c(\mathcal{F}_c(\tilde{\Phi}_c)) + L_f(\mathcal{F}_f(\tilde{\Phi}_f))}_{\text{Distortion}} + \lambda \underbrace{\sum_{\phi \in \Phi} I(\phi)}_{\text{Rate}} \tag{2}$$

where $\Psi$ denotes the parameters of $\mathcal{F}$ as well any parameters in the prior distribution $q$, and we have explicitly split $\Phi$ into the coarse $\Phi_c$ and fine $\Phi_f$ components such that $\Phi = \{\Phi_c, \Phi_f\}$. $\lambda$ is a trade-off parameter that balances between rate and distortion. A rate–distortion (RD) plot can be traced by varying $\lambda$ to explore the performance of the compressed model at different bitrates.

**Compressing a single scene** When training cNeRF to render a single scene, we have to choose how to parameterize and structure $\mathcal{F}$ and the prior distribution $q$ over the network weights. Since the networks are MLPs, the model parameters for a layer $l$ consist of the kernel weights and biases $\{W_l, b_l\}$. We compress only the kernel weights $W_l$, leaving the bias uncompressed since it is much smaller in size. The quantized kernel weights $\tilde{W}_l$ are mapped to the model weights by $\mathcal{F}_l$, i.e. $W_l = \mathcal{F}_l(\tilde{W}_l)$. $\mathcal{F}_l$ is constructed as an affine scalar transformation, which is applied elementwise to $\tilde{W}_l$:

$$\mathcal{F}_l(\tilde{W}_{l,ij}) = \alpha_l \tilde{W}_{l,ij} + \beta_l \tag{3}$$

We take the prior to be factored over the layers, such that we learn a prior per linear kernel $q_l$. Within each kernel, we take the weights in $\tilde{W}_l$ to be i.i.d. from the univariate distribution $q_l$, parameterized by a small MLP, as per the approach of Ballé et al. (2017). Note that the parameters of this MLP can be discarded after training (once the probability mass functions have been built).

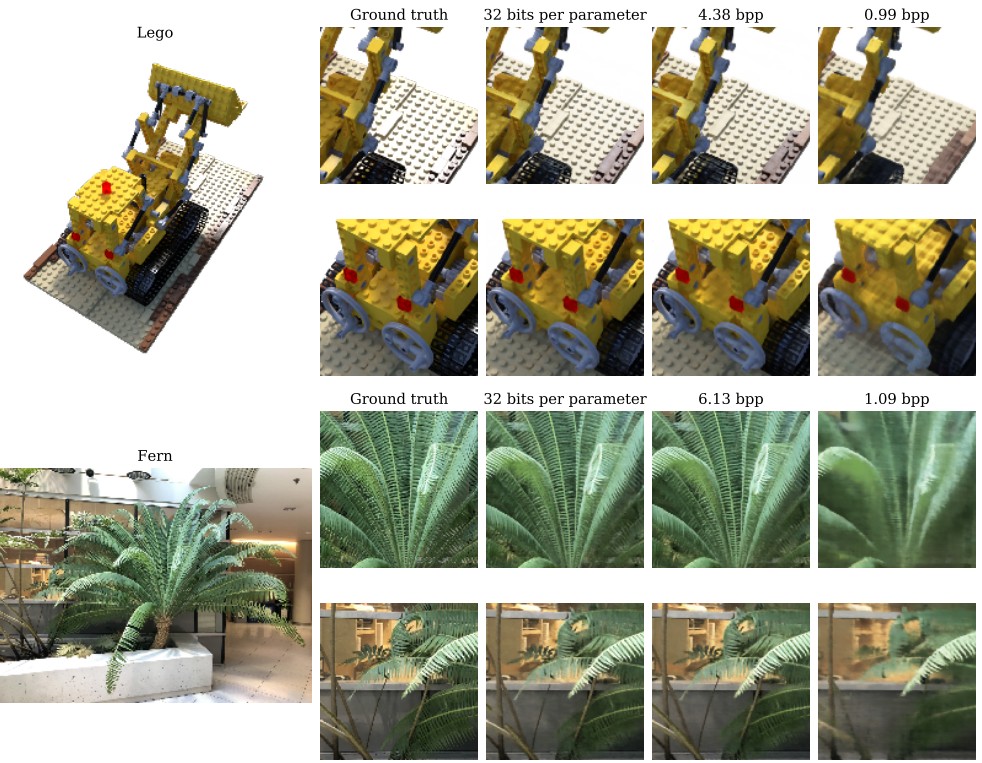

Figure 2: Renderings of the synthetic Lego scene and real Fern scene from the uncompressed NeRF model, at 32 bits per parameter (bpp), and from cNeRF with $\lambda \in \{0.0001, 0.01\}$.

**Compressing multiple scenes** While the original NeRF model is trained for a single scene, we hypothesize that better rate–distortion performance can be achieved for multiple scenes, especially if they share information, by training a joint model. For a dataset of $M$ scenes, we parameterize the kernel weights of model $m$, layer $l$ as:

$$W_l^m = \mathcal{F}_l^m(\tilde{W}_l^m, \tilde{S}_l)$$
$$= \alpha_l^m \tilde{W}_l^m + \beta_l^m + \gamma_l \tilde{S}_l \qquad (4)$$

Compared to Eqn. 3, we have added a shift, parameterized as a scalar linear transformation of a discrete shift $\tilde{S}_l$, that is shared across all models $m \in \{1, ..., M\}$. $\tilde{S}_l$ has the same dimensions as the kernel $W_l^m$, and as with the discrete latent kernels, $\tilde{S}_l$ is coded by a learned probability distribution. The objective for the multi-scene model becomes:

$$\mathcal{L}(\Phi, \Psi) = \sum_{m=1}^{M} \left[ L_c^m(\mathcal{F}_c^m(\tilde{\Phi}_c^m, \tilde{\Phi}_c^s)) + L_f^m(\mathcal{F}_f^m(\tilde{\Phi}_f^m, \tilde{\Phi}_f^s)) + \lambda \sum_{\phi \in \Phi^m} I(\phi) \right] + \lambda \sum_{\phi \in \Phi^s} I(\phi) \quad (5)$$

where $\Phi^s$ is the set of all discrete shift $\tilde{S}$ parameters, and the losses, latent weights and affine transforms are indexed by scene and model $m$. Note that this parameterization has *more* parameters than the total of the $M$ single scene models, which at first appears counter-intuitive, since we wish to reduce the overall model size. It is constructed as such so that the multi-scene parameterization contains the $M$ single scene parameterizations - they can be recovered by setting the shared shifts to zero. If the shifts are set to zero then their associated probability distributions can collapse to place all their mass at zero. So we expect that if there is little benefit to using the shared shifts then they can be effectively ignored, but if there is a benefit to using them then they can be utilized. As such, we can interpret this parameterization as inducing a soft form of parameter sharing.

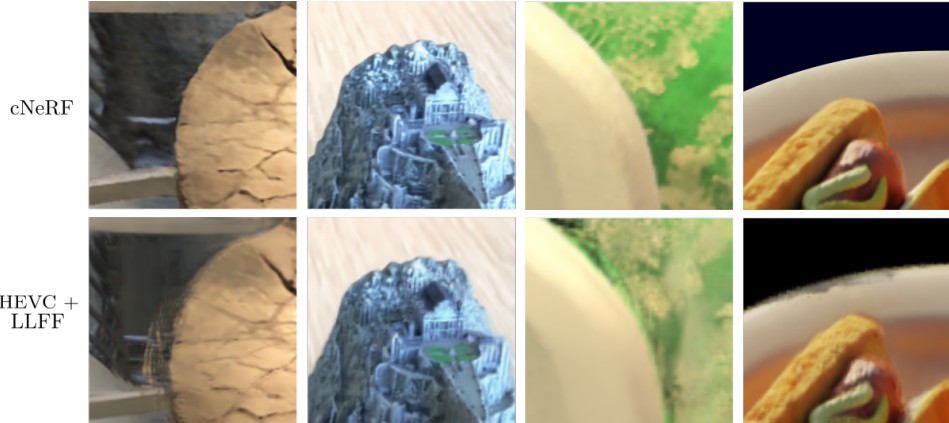

Figure 3: A comparison of four (zoomed in) renderings from cNeRF with $\lambda = 0.0001$ and HEVC + LLFF with QP=30. HEVC + LLFF shows obvious artifacts such as ghosting around edges and an overall less crisp rendering.

## 4 EXPERIMENTS

**Datasets**  To demonstrate the effectiveness of our method, we evaluate on two sets of scenes used by Mildenhall et al. (2020):

- *Synthetic*. Consisting of $800 \times 800$ pixel$^2$ views taken from either the upper hemisphere or entire sphere around an object rendered using the Blender software package. There are 100 views taken to be in the train set and 200 in the test set.

- *Real*. Consisting of a set of forward facing $1008 \times 756$ pixel$^2$ photos of a complex scene. The number of images varies per scene, with $1/8$ of the images taken as the test images.

Since we are interested in the ability of the receiver to render novel views, all distortion results (for any choice of perceptual metric) presented are given on the test sets.

**Architecture and Optimization**  We maintain the same architecture for the NeRF model as Mildenhall et al. (2020), consisting of 13 linear layers and ReLU activations. For cNeRF we use Adam (Kingma & Ba, 2015) to optimize the latent weights $\Phi$ and the weights contained in the decoding functions $\mathcal{F}$. For these parameters we use initial learning rate of $5 \times 10^{-4}$ and a learning rate decay over the course of learning, as per Mildenhall et al. (2020). For the parameters of the learned probability distributions $q$, we find it beneficial to use a lower learning rate of $5 \times 10^{-5}$, such that the distributions do not collapse prematurely. We initialize the latent linear kernels using the scheme of Glorot & Bengio (2010), the decoders $\mathcal{F}$ near the identity.

**Baseline**  We follow the general methodology exhibited in light field compression and take the compressed representation of the scene to be a compressed subset of the views. The receiver then decodes these views, and renders novel views conditioned on the reconstructed subset. We use the video codec HEVC to compress the subset of views, as is done by Jiang et al. (2017). To render novel views conditioned on the reconstructed set of views, we choose the Local Light Field Fusion (LLFF) approach of Mildenhall et al. (2019). LLFF is a state-of-the-art learned approach in which a novel view is rendered by promoting nearby views to multiplane images, which are then blended. We refer to the full baseline subsequently as HEVC + LLFF.

### 4.1 RESULTS

**Single scene compression**  To explore the frontier of achievable rate–distortion points for cNeRF, we evaluate at a range of entropy weights $\lambda$ for four scenes – two synthetic (Lego and Ficus) and two real (Fern and Room). To explore the rate–distortion frontier for the HEVC + LLFF baseline we evaluate at a range of QP values for HEVC. We give a more thorough description of the exact specifications of the HEVC + LLFF baseline and the ablations we perform to select the hyperparameter values in Appendix Sec. B. We show the results in Fig. 4. We also plot the performance of

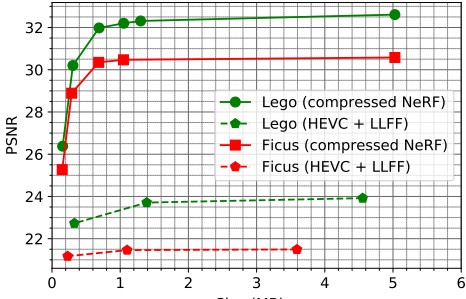 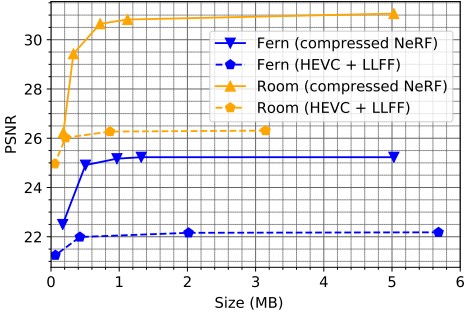

Figure 4: Rate–distortion curves for both the cNeRF and HEVC + LLFF approaches, on two real (left) and two synthetic (right) scenes. We truncate the curves for HEVC + LLFF, since increasing the bitrate further does not improve PSNR. See Fig. 7 for the full curves.

| | **NeRF** | **cNeRF** ($\lambda = 1e^{-4}$) | | | **HEVC + LLFF** (QP=30) | |
|---|---|---|---|---|---|---|
| Scene | PSNR | PSNR | Size (KB) | Reduction | PSNR | Size (KB) |
| Chair | 33.51 | 32.28 | 621 | 8.32× | 24.38 | 5,778 |
| Drums | 24.85 | 24.85 | 870 | 5.94× | 20.25 | 5,985 |
| Ficus | 30.58 | 30.35 | 701 | 7.37× | 21.49 | 3,678 |
| Hotdog | 35.82 | 34.95 | 916 | 5.65× | 30.18 | 2,767 |
| Lego | 32.61 | 31.98 | 707 | 7.31× | 23.92 | 4,665 |
| Materials | 29.71 | 29.17 | 670 | 7.71× | 22.49 | 3,134 |
| Mic | 33.68 | 32.11 | 560 | 9.23× | 28.95 | 3,032 |
| Ship | 28.51 | 28.24 | 717 | 7.21× | 24.95 | 5,881 |
| Room | 31.06 | 30.65 | 739 | 7.00× | 26.27 | 886 |
| Fern | 25.23 | 25.17 | 990 | 5.22× | 22.16 | 2,066 |
| Leaves | 21.10 | 20.95 | 1,154 | 4.48× | 18.15 | 3,162 |
| Fortress | 31.65 | 31.15 | 818 | 6.32× | 26.57 | 1,149 |
| Orchids | 20.18 | 20.09 | 1,218 | 4.24× | 17.87 | 2,357 |
| Flower | 27.42 | 27.21 | 938 | 5.51× | 23.46 | 1,009 |
| T-Rex | 27.24 | 26.72 | 990 | 5.22× | 22.30 | 1,933 |
| Horns | 27.80 | 27.28 | 995 | 5.20× | 20.71 | 2,002 |

Table 1: Results comparing the uncompressed NeRF model, cNeRF and HEVC + LLFF baseline. We pick $\lambda$ and QP to give a reasonable trade-off between bitrate and PSNR. The reduction column is the reduction in the size of cNeRF as compared to the uncompressed NeRF model, which has a size of 5,169KB. Note that for all scenes, cNeRF achieves both a higher PSNR and a lower bitrate than HEVC + LLFF.

the uncompressed NeRF model – demonstrating that by using entropy penalization the model size can be reduced substantially with a relatively small increase in distortion. For these scenes we plot renderings at varying levels of compression in Fig. 2 and Fig. 8. The visual quality of the renderings does not noticeably degrade when compressing the NeRF model down to bitrates of roughly 5-6 bits per parameter (the precise bitrate depends on the scene). At roughly 1 bit per parameter, the visual quality has degraded significantly, although the renderings are still sensible and easily recognisable. We find this to be a surprising positive result, given that assigning a single bit per parameter is extremely restrictive for such a complex regression task as rendering. Indeed, to our knowledge no binary neural networks have been demonstrated to be effective on such tasks.

Although the decoding functions $\mathcal{F}$ (Eqn. 3) are just relatively simple scalar affine transformations, we do not find any benefit to using more complex decoding functions. With the parameterization given, most of the total description length of the model is in the coded latent weights, not the parameters of the decoders or entropy models. We give a full breakdown in Tab. 5.

Fig. 4 shows that cNeRF clearly outperforms the HEVC + LLFF baseline, always achieving lower distortions at a (roughly) equivalent bitrate. Reconstruction quality is reported as peak signal-to-noise ratios (PSNR). The results are consistent with earlier demonstrations that NeRF produces much better renderings than the LLFF model (Mildenhall et al., 2020). However, it is still interesting

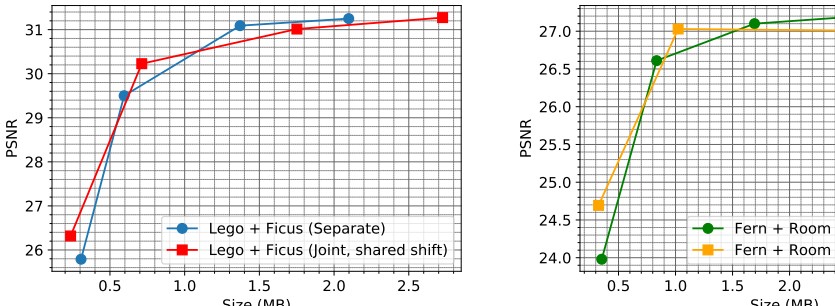

Figure 5: Rate–distortion curves for comparing the multi-scene model with a single shared shift to the single scene models. The models are shown for two synthetic (left) and two real scenes (right).

to see that this difference persists even at much lower bitrates. To evaluate on the remaining scenes, we select a single $\lambda$ value for cNeRF and QP value for HEVC + LLFF. We pick the values to demonstrate a reasonable trade-off between rate and distortion. The results are shown in Tab. 1. For every scene the evaluated approaches verify that cNeRF achieves a lower distortion at a lower bitrate. We can see also that cNeRF is consistently able to reduce the model size significantly without seriously impacting the distortion. Further, we evaluate the performance of cNeRF and HEVC + LLFF for other perceptual quality metrics in Tab. 3 and 4. Although cNeRF is trained to minimize the squared error between renderings and the true images (and therefore maximize PSNR), cNeRF also outperforms HEVC + LLFF in both MS-SSIM (Wang et al., 2003) and LPIPS (Zhang et al., 2018). This is significant, since the results of Mildenhall et al. (2020) indicated that for SSIM and LPIPS, the LLFF model had a similar performance to NeRF when applied to the real scenes. We display a comparison of renderings from cNeRF and HEVC + LLFF in Fig. 3.

**Multi-scene compression** For the multi-scene case we compress one pair of synthetic scenes and one pair of real scenes. We train the multi-scene cNeRF using a single shared shift per linear kernel, as per Eqn. 4. To compare the results to the single scene models, we take the two corresponding single scene cNeRFs, sum the sizes and average the distortions. We plot the resulting rate–distortion frontiers in Fig. 5. The results demonstrate that the multi-scene cNeRF improves upon the single scene cNeRFs at low bitrates, achieving higher PSNR values with a smaller model. This meets our expectation, since the multi-scene cNeRF can share parameters via the shifts (Eqn. 4) and so decrease the code length of the scene-specific parameters. At higher bitrates we see no benefit to using the multi-scene parameterization, and in fact see slightly worse performance. This indicates that in the unconstrained rate setting, there is no benefit to using the shared shifts, and that they may slightly harm optimization.

## 5 RELATED WORK

**Scene Compression** A 3D scene is typically represented as a set of images, one for each view. For a large number of views, compressing each image individually using a conventional compression method can require a large amount of space. As a result, there is a body of compression research which aims to exploit the underlying scene structure of the 3D scene to reduce space requirements. A lot of research has been focused on compressing light field image (LFI) data (Astola & Tabus, 2018; Jiang et al., 2017; Bakir et al., 2018; Jia et al., 2019; Zhao et al., 2018). LFI data generally consists of multiple views with small angular distances separating them. This set of views can be used to reconstruct a signal on the 4D domain of rays of the light field itself, thus permitting post-processing tasks such as novel view synthesis and refocusing. A majority of works select a representative subset of views to transmit from the scene. These are compressed and transmitted, typically using a video codec, with the receiver decoding these images and then rendering any novel view for an unobserved (during training) camera pose. Reconstruction for novel camera poses can be performed using traditional methods, such as optical flow (Jiang et al., 2017), or by using recent learned methods that employ convolutional neural networks (Zhao et al., 2018) and generative adversarial networks (Jia et al., 2019). A contrasting approach to multi-view image compression is proposed by Liu et al. (2019), in which a pair of images from two viewpoints is compressed by con-

ditioning the coder of the second image on the coder of the first image. It is important to emphasise that we are not studying this kind of approach in this work, since we wish the receiver to have the ability to render novel views.

**Neural Rendering** is an emerging research area which combines learned components with rendering knowledge from computer graphics. Recent work has shown that neural rendering techniques can generate high quality novel views of a wide range of scenes (Mildenhall et al., 2020; Sitzmann et al., 2019; Liu et al., 2020; Schwarz et al., 2020). In this work we build upon the method of Mildenhall et al. (2020), coined as a Neural Radiance Field (NeRF), for single scene compression and then extend it with a novel reparameterization for jointly compressing multiple scenes. Training neural representation networks jointly across different scenes (without compression) has been explored by Sitzmann et al. (2019) and Liu et al. (2020), who use a hypernetwork (Ha et al., 2017) to map a latent vector associated with each scene to the parameters of the representation network. Liu et al. (2020) note that the hypernetwork approach results in significant degradation of performance when applied to the NeRF model (a loss of more than 4 dB PSNR). In contrast, our approach of shared reparameterization is significantly different from these methods.

**Model Compression** There is a body of research for reducing the space requirements of deep neural networks. Pruning tries to find a sparse set of weights by successively removing a subset of weights according to some criterion (Han et al., 2016; Li et al., 2017). Quantization reduces the precision used to describe the weights themselves (Courbariaux et al., 2016; Li et al., 2016). In this work we focus instead on weight coding approaches (Havasi et al., 2019; Oktay et al., 2020) that code the model parameters to yield a compressed representation.

## 6 DISCUSSION AND CONCLUSION

Our results demonstrate that cNeRF produces far better results as a compressed representation than a state-of-the-art baseline, HEVC+LLFF, which follows the paradigm of compressing the original views. In contrast, our method compresses a representation of the radiance field itself. This is important for two reasons:

- Practically, compressing the views themselves bars the receiver from using more complex and better-performing rendering methods such as NeRF, because doing this would require training to be performed at the receiving side after decompression, which is computationally infeasible in many applications.

- Determining the radiance field and compressing it on the sending side may have coding and/or representational benefits, because of the data processing inequality: the cNeRF parameters are a function of the original views, and as such must contain equal to or less information than the original views (the training data). The method is thus relieved of the need to encode information in the original views that is not useful for the rendering task.

It is difficult to gather direct evidence for the latter point, as the actual entropy of both representations is difficult to measure (we can only upper bound it by the compressed size). However, the substantial performance improvement of our method compared to HEVC+LLFF suggests that the radiance field is a more economical representation for the scene.

Linked to our choice is also the fact that we adopt a more realistic evaluation methodology than many scene compression techniques. Rather reporting the bitrate and reconstruction quality of the original views, we evaluate our method (and the baseline) by reporting the reconstruction quality of a held-out set of views of each scene, which was not used for training. Since in a free-viewpoint scenario, the vast majority of rendered views will not correspond to one of the original ones, we believe this more accurately measures success of the compared methods.

The encoding time for cNeRF is long, given that a new scene must be trained from scratch. Importantly though, the decoding time is much less, as it is only required to render the views using the decompressed NeRF model. cNeRF enables neural scene rendering methods such as NeRF to be used for scene compression, as it shifts the complexity requirements from the receiver to the sender. In many applications, it is more acceptable to incur high encoding times than high decoding times, as one compressed data point may be decompressed many times, allowing amortization of the encoding time, and since power-constrained devices are often at the receiving side. Thus, our method represents a big step towards enabling neural scene rendering in practical applications.

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

# A    NEURAL RADIANCE FIELDS

The neural rendering approach of Mildenhall et al. (2020) uses a neural network to model a radiance field. The radiance field itself is a learned mapping $g_\theta : \mathbb{R}^5 \to (\mathbb{R}^3, \mathbb{R}^+)$, where the input is a 3D spatial coordinate $\boldsymbol{p} = (x, y, z) \in \mathbb{R}^3$ and a 2D viewing direction $\boldsymbol{d} = (\theta, \phi) \in \mathbb{R}^2$. The NeRF model also makes use of a positional encoding into the frequency domain, applied elementwise to spatial and directional inputs

$$\gamma(p) = (\sin(2^0 \pi p), \cos(2^0 \pi p), ..., \sin(2^{L-1} \pi p), \cos(2^{L-1} \pi p)) \tag{6}$$

This type of encoding has been shown to be important for implicit models, which take as input low dimensional data which contains high frequency information (Tancik et al., 2020; Sitzmann et al., 2020).

The network output is an RGB value $\boldsymbol{c} = (r, g, b) \in \mathbb{R}^3$ and a density element $\sigma \in \mathbb{R}^+$. To render a particular view, the RGB values are sampled along the relevant rays and accumulated according to their density elements. In particular, the color $\boldsymbol{c}(\boldsymbol{r})$ of a ray $\boldsymbol{r} = \{\boldsymbol{o} + t\boldsymbol{d} : t \geq 0\}$, in direction $\boldsymbol{d}$ from the camera origin $\boldsymbol{o}$, is computed as

$$\boldsymbol{c}(\boldsymbol{r}) = \sum_{i=1}^{K} T_i(1 - \exp(-\sigma_i \delta_i))\boldsymbol{c}_i, \quad \text{where } T_i = \exp\left(-\textstyle\sum_{j=1}^{i-1} \sigma_j \delta_j\right), \tag{7}$$

where $(\boldsymbol{c}_i, \sigma_i)$ is the output of the mapping evaluated at $(\boldsymbol{p}_i, \boldsymbol{d})$, where $\boldsymbol{p}_i = \boldsymbol{o} + t_i \boldsymbol{d}$, $t_i$ is the distance of sample $i$ from the origin along the ray, and $\delta_i = t_{i+1} - t_i$ is the distance between samples. The color $\boldsymbol{c}(\boldsymbol{r})$ can be interpreted as the expected color of the point along the ray in the scene closest to the camera, if the points in the scene are distributed along the ray according to an inhomogeneous Poisson process. Since in a Poisson process with density $\sigma_i$, the probability that there are no points in an interval of length $\delta_i$ is $\exp(-\sigma_i \delta_i)$. Thus $T_i$ is the probability that there are no points between $t_1$ and $t_i$, and $(1 - \exp(-\sigma_i \delta_i))$ is the probability that there is a point between $t_i$ and $t_{i+1}$. The rendered view $\hat{X}$ comprises pixels whose colors $\boldsymbol{c}(\boldsymbol{r})$ are evaluated at rays emanating from the same camera origin $\boldsymbol{o}$ but having slightly different directions $\boldsymbol{d}$, depending on the camera pose $V$.

# B    HEVC + LLFF SPECIFICATION AND ABLATIONS

There are many hyperparameters to select for the HEVC + LLFF baseline. The first we consider is the number of images to compress with HEVC. If too many images are compressed with HEVC then at some point the performance of LLFF will saturate and an unnecessary amount of space will be used. On the other hand, if too few images are compressed with HEVC, then LLFF will find it difficult to blend these (de)compressed images to form high quality renders. To illustrate this effect, we run an ablation on the Fern scene where we vary the number of images we compress with HEVC, rendering a held out set of images conditioned on the reconstructions. The results are displayed in Fig. 6. We can clearly see the saturation point at around 10 images, beyond which there is no benefit to compressing extra images. Thus when picking the number of images to compress for new scenes, we do not use more than 4 per test image (which corresponds to compressing 12 images in our ablation).

The second effect we study is the order in which images are compressed with HEVC, which affects the performance as HEVC is a video codec and thus sensitive to image ordering. It stands to reason that the more the sequence of images resemble a natural video, the better coding will be. As such, we consider two orderings: firstly the "snake scan" ordering, in which images are ordered vertically by their camera pose, going alternately left to right then right to left. The second is the "lozenge" ordering (Jiang et al., 2017), in which images are ordered by the camera pose in a spiral outwards from their centre. Both orderings appear sensible since they always step from a given camera pose to an adjacent pose. We compare results compressing and reconstructing a set of images using HEVC across a range of Quantization Parameter (QP) values for the Fern scene in Tab. 2. The difference between the two orderings is very small. Since snake scan is simpler to implement, we use this in all our experiments.

The effect of changing QP is demonstrated in Fig. 7, and we select QP=30 for the experiments in which we choose one rate–distortion point to evaluate, since it achieves almost the same performance as QP=20 and QP=10 with considerably less space.

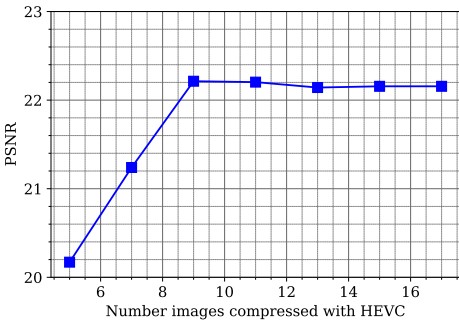

Figure 6: Test performance of the HEVC + LLFF baseline across different number of images compressed with HEVC. The decompressed images are used by LLFF to reconstruct the test views.

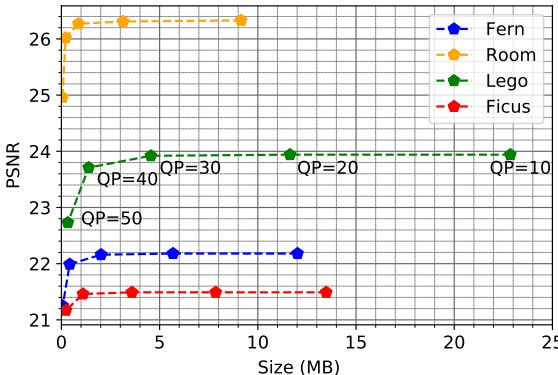

Figure 7: Full rate–distortion curves for HEVC + LLFF, with labels showing the effect of the QP parameter. To avoid clutter, only the Lego QP labels are given, and the other scenes are similarly ordered from QP=10 on the right to QP=50 on the left.

## C    EXTRA RESULTS

Here we present some further results from our experiments, including results on different perception metrics, a breakdown of the cNeRF model size and extra comparisons of renderings.

| QP | Snake scan | Lozenge |
|----|-----------|---------|
| 10 | 50.9 | 50.8 |
| 20 | 42.5 | 42.4 |
| 30 | 33.9 | 33.8 |
| 40 | 26.7 | 26.8 |
| 50 | 22.6 | 22.5 |

Table 2: PSNR values from compressing and reconstructing a set of images from the Fern scene, with two different orderings on the images.

| | cNeRF ($\lambda = 1e^{-4}$) | | HEVC + LLFF (QP=30) | |
|---|---|---|---|---|
| Scene | PSNR(Y) | PSNR(UV) | PSNR(Y) | PSNR(UV) |
| Chair | 36.64 | 45.21 | 33.41 | 42.46 |
| Drums | 26.68 | 37.17 | 21.72 | 34.72 |
| Ficus | 31.77 | 43.50 | 24.05 | 37.00 |
| Hotdog | 36.31 | 42.70 | 31.98 | 41.62 |
| Lego | 29.89 | 40.56 | 24.94 | 35.92 |
| Materials | 26.82 | 38.53 | 16.23 | 35.04 |
| Mic | 33.36 | 48.02 | 29.87 | 48.64 |
| Ship | 28.27 | 38.46 | 26.48 | 38.48 |
| Room | 32.59 | 44.94 | 27.05 | 42.50 |
| Fern | 25.16 | 40.33 | 22.15 | 38.06 |
| Leaves | 21.17 | 36.09 | 18.43 | 34.52 |
| Fortress | 31.60 | 44.70 | 27.31 | 41.67 |
| Orchids | 20.43 | 34.65 | 18.17 | 32.11 |
| Flower | 27.91 | 38.25 | 24.27 | 33.90 |
| T-Rex | 26.77 | 42.64 | 22.42 | 40.07 |
| Horns | 27.68 | 42.65 | 22.57 | 40.13 |

Table 3: PSNR values comparing cNeRF to HEVC + LLFF. The images are rendered in the YUV color encoding and the PSNR is computed in the Y channel and average of the UV channels. cNeRF is superior to HEVC + LLFF in PSNR(Y) and PSNR(UV) for all scenes except PSNR(UV) for Mic and Ship. Note that the bitrates are the same as for Tab. 1.

| | cNeRF ($\lambda = 1e^{-4}$) | | | HEVC + LLFF (QP=30) | | |
|---|---|---|---|---|---|---|
| Scene | MS-SSIM(Y) | MS-SSIM(RGB) | LPIPS | MS-SSIM(Y) | MS-SSIM(RGB) | LPIPS |
| Chair | 0.997 | 0.997 | 0.014 | 0.989 | 0.989 | 0.026 |
| Drums | 0.953 | 0.952 | 0.070 | 0.910 | 0.909 | 0.109 |
| Ficus | 0.986 | 0.986 | 0.023 | 0.925 | 0.924 | 0.069 |
| Hotdog | 0.992 | 0.990 | 0.041 | 0.983 | 0.981 | 0.070 |
| Lego | 0.983 | 0.980 | 0.034 | 0.951 | 0.945 | 0.080 |
| Materials | 0.971 | 0.970 | 0.047 | 0.833 | 0.832 | 0.172 |
| Mic | 0.992 | 0.992 | 0.022 | 0.988 | 0.988 | 0.026 |
| Ship | 0.891 | 0.889 | 0.201 | 0.879 | 0.883 | 0.174 |
| Room | 0.979 | 0.977 | 0.087 | 0.950 | 0.948 | 0.168 |
| Fern | 0.934 | 0.932 | 0.187 | 0.862 | 0.862 | 0.239 |
| Leaves | 0.909 | 0.905 | 0.204 | 0.843 | 0.842 | 0.244 |
| Fortress | 0.966 | 0.962 | 0.090 | 0.918 | 0.914 | 0.165 |
| Orchids | 0.852 | 0.849 | 0.220 | 0.720 | 0.721 | 0.320 |
| Flower | 0.945 | 0.941 | 0.138 | 0.911 | 0.906 | 0.171 |
| T-Rex | 0.962 | 0.961 | 0.101 | 0.905 | 0.903 | 0.182 |
| Horns | 0.953 | 0.951 | 0.169 | 0.886 | 0.884 | 0.238 |

Table 4: MS-SSIM and LPIPS values comparing cNeRF to HEVC + LLFF. MS-SSIM is given in both the Y channel (from the YUV color encoding) and RGB. LPIPS is given for RGB, as it is only defined for such. In all cases, cNeRF is superior to HEVC + LLFF in MS-SSIM and LPIPS. Note that the bitrates are the same as for Tab. 1

| Entropy weight $\lambda$ | Rate (KB) | Overhead (KB) |
|---|---|---|
| $1 \times 10^{-2}$ | 119 | 23 |
| $1 \times 10^{-3}$ | 293 | 27 |
| $1 \times 10^{-4}$ | 673 | 34 |
| $1 \times 10^{-5}$ | 1061 | 42 |

Table 5: Breakdown of the cNeRF size across four entropy weights trained on the Lego scene. The size is divided into the size of the coded latent weights (the rate) and everything else (the overhead). The overhead consists of description lengths of the probability built from the prior $q$, the parameters of the decoding functions $\mathcal{F}$ and any bias parameters.

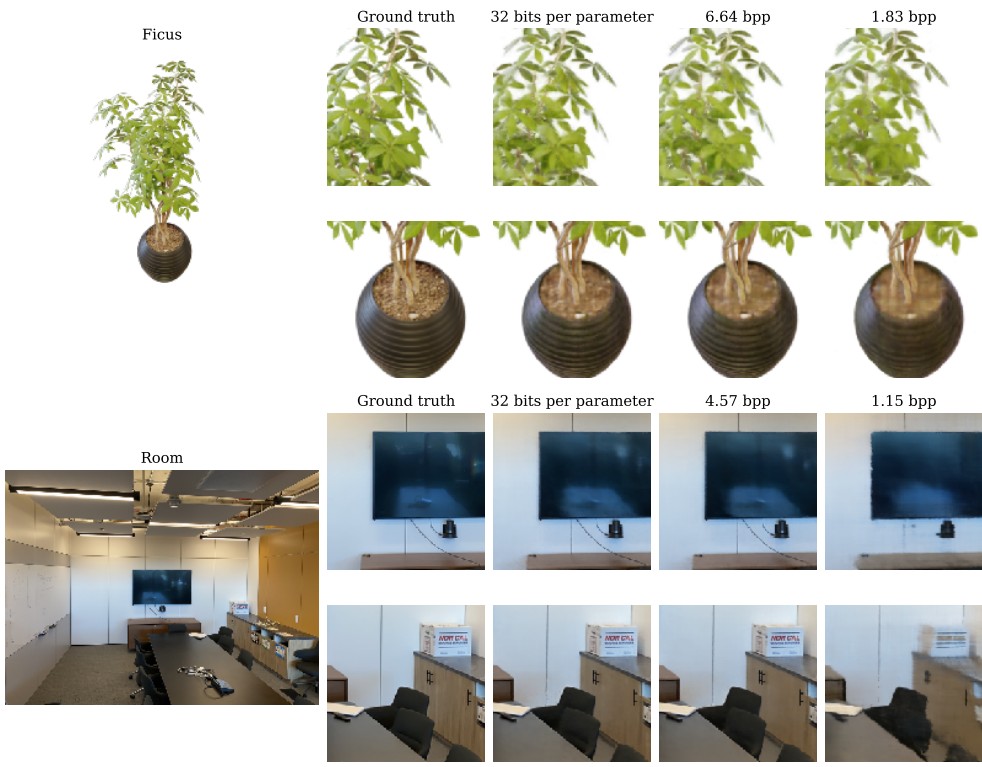

Figure 8: Renderings of the synthetic Ficus scene and real Room scene from the uncompressed NeRF model, at 32 bits per parameter (bpp), and from cNeRF with $\lambda \in \{0.0001, 0.01\}$.