# OpenReview forum: "3D Scene Compression through Entropy  Penalized Neural Representation Functions"
_ICLR.cc/2021/Conference — Reject_

### Official Review · AnonReviewer4 · 2020-10-24
**3D Scene Compression through Entropy Penalized Neural Representation Functions**

**Rating:** 5
**Confidence:** 4

**Review:**

I think the paper is well written, and explains the details of the method well. I addition I think choices made in the method are intelligent and well justified.

My concerns with the paper lie in two area. The first is that I am not convinced that the problem this paper seeks to solve, ie  compression of a 3D scene, is so relevant to the ML research community that is justifies the lack of novelty in the method. From reading this paper it feel like 2 approaches have been found , ie, NeRF and model compression using entropy penalty, which happen to work well together, but the actual degree of novel research contribution seems low.  If 3D scene compression was an established area of research in the ML research community, with many previous works proposing high performing solutions,  this approach may be justified as sufficiently better across various metrics. However, given the lack of this prior work, the solution feels far more like an engineering solution to allow NeRF to work well on cellphones then machine learning research.

My second concern is the lack of comparison to other approaches to scene compression. From the way I view the problem you have proposed, you are assuming practically explicit 3D scene information as input and attempting to transfer this information between devices in as small a size as possible such that it images of the scene can be sampled on the new device in as high a quality as possible.  There are other ways of describing a 3D scene then with multiple images. The experiment I really want to see here is if this method is better then if the explicit scene information is compressed and rendered on the second device. If you can provide this in new experiments I will be happy to raise my score. Examples of this would be compressing  a 3D mesh, or surface defined by voxels with lighting parameters and textures.  My main concern here is that for simple scenes I am very sure that compressing or even transferring directly the 3D scene explicitly is a far superior, though there is probably some level of scene complexity at which your method overtakes it. I think it is important to establish this, and understand where your method is applicable.

---

> ### Author Response · Authors · 2020-11-24
> **Response**
>
> We thank the reviewer for their thoughtful comments. We provide responses below:
>
> *The first is that I am not convinced that the problem this paper seeks to solve, ie compression of a 3D scene, is so relevant to the ML research community*
>
> The concept of compressing a representation of a 3d scene is relatively new, and the set of applications for such methods has not yet been well established. However, we do believe there is real value in the research of baseline methods which can be used to do 3d scene compression, given the clear emergence of the technology to create and consume 3d visual data.
>
> *My second concern is the lack of comparison to other approaches to scene compression... Examples of this would be compressing a 3D mesh, or surface defined by voxels with lighting parameters and textures*
>
> We do accept that it will most likely be more efficient to transmit compressed meshes if the scene itself is easily described in this way, and there is a body of literature that proposes such methods (through for example the transmission of truncated signed distance fields). However, our method works equally for these synthetic scenes (which are easily described by meshes), and for natural scenes, which are far more complex. Sending a compressed representation of a natural scene is not feasible using the methods you propose.  In particular, meshes, voxels, and other traditional geometric representations cannot generally capture the view-dependent effects present in natural scenes.  This is why we do not use it as a method for comparison.
>
> Given that we wish to compare to scene compression methods that can model both synthetic and natural scenes, the pool of methods that can do so shrinks considerably. We believe that we have used a method (HEVC + LLFF) which is a fair comparison, and it is directly inspired by methods within the field of light-field image (LFI) compression. This is a relatively well-studied topic, and which is trying to solve a problem that is a subset of the problem we are trying to solve, since LFI are views of a 3d scene with a small angular distance between views.

---

### Official Review · AnonReviewer3 · 2020-10-28

**Rating:** 5
**Confidence:** 2

**Review:**

Summary:

This paper proposes to compress nerf models with entropy loss, where instead of directly training nerf model parameters, it trains a new function F which takes some compressed information and decodes to the nerf models. Then it did the same things as nerf, which render scenes in novel views. The authors show that the function F could largely compress the original nerf models while keeping similar PSNR.

Comments:
The paper combines network compression and neural renderings, which is pretty interesting. However, I have several concerns :

Novelty:  The paper seems to combine two methods together, where in network compression, it adopts Oktay et al. (2020) while in neural renderings, it relies on nerf. Though it introduces compression to neural rendering, such a combination seems to be not very creative.

Generalization. While one can train such a network for nerf scenes, the network may get overfit to these scenes since the whole dataset has no more than 20 scenes.  It would be great to validate on shapenet datasets, or some dataset has at least 100 models.

Comaprison. The authors compare their methods with HEVC+LLFF method, which is pretty unfair. Since HEVS is a traditional video compression method while LIFF is also a traditional blending view rendering method. The authors are encouraged to compare with some neural network compression methods + neural rendering methods.

Conclusion: Overall, I think this paper proposes an interesting idea and shows good results. However, due to lack of creativity and unfair comparison, I rate it below the acceptance bar.

---

> ### Author Response · Authors · 2020-11-24
> **Response**
>
> We thank the reviewer for their comments. Find our responses below:
>
> *While one can train such a network for nerf scenes, the network may get overfit to these scenes since the whole dataset has no more than 20 scenes.*
>
> The NeRF model is in fact just trained for a single scene - there is no generalization, except in the multi-scene approach where we learn jointly across scenes. Training a NeRF model for a wide range of scenes is an open problem in the research community.
>
> The decompressor for the (compressed) NeRF model, on the other hand, does not depend on any dataset.  Its parameters, which include scales and offsets applied to the integer latent variables, and which also include the probabilities used to entropy code the integer latents, are all transmitted to the decoder in a fixed-length (e.g., 16-bit) integer format, without need for any training, thus obviously generalizing to any data.
>
> *The authors compare their methods with HEVC+LLFF method, which is pretty unfair. Since HEVS is a traditional video compression method while LIFF is also a traditional blending view rendering method. The authors are encouraged to compare with some neural network compression methods + neural rendering methods.*
>
> LLFF is not a traditional blending approach - a neural network is trained to promote images to multi-plane images, which are then blended. It is a state-of-the-art, learned method for novel view synthesis. HEVC is not a learned approach, but is still a modern and widely used video codec. Given that neural network compression approaches for video are still in their infancy, there is no obvious neural compression codec to use. As such, we think that HEVC + LLFF is a strong and appropriate baseline, which gives the best chance to this general methodology of communicating a compressed set of views, and rendering any novel views conditioned on those.

---

### Official Review · AnonReviewer2 · 2020-10-28
**Interesting problem, but not enough novelty and the not well-conducted experiements**

**Rating:** 4
**Confidence:** 5

**Review:**

**Paper Summary**:

Compressing 3D scene is an interesting problem to explore. The paper proposes to add entropy penalized reparametrization (Oktay et al. (2020)) technique into Nerf (Mildenhall et al. (2020)) and compress the neural network in Nerf. Experiments also show some compressing rates improvement with the proposed baseline, which I have some concerns in the weakness below.

**Strength**:
1. The problem this paper is attacking is interesting, compressing 3D scene plays an important role in real-time applications.
2. The paper also showed the approach to compress multiple scenes in one network (though I have some concerns for the experiments in weakness below).

**Weakness**:
1. Lack of novelty. The main technique from this paper is merging two methods together (Oktay et al. (2020), and Mildenhall et al. (2020)),  with some improvements by extending the nerf to multiple scenes. However, the multiple scenes experiments are not performed well to demonstrate the effectiveness of the proposed method (see below), so the overall novelty is not enough.

2. Experiments are not great enough to show the effectiveness of the full pipeline.

   a) In single scene experiment, I think the main reason why HEVC+LLFF has lower PSNR is that LLFF is interpolating multiple training views, and this is obvious and has already been shown in the original Nerf paper (Table 1, Fig 5 in  Mildenhall et al. (2020)). Therefore, this is not a valid experiment to show the pipeline in this paper is better in compression. A more valid baseline should be: receiver receives the training images compressed by HEVC -> decode the images -> receiver train a new Nerf on the decoded images -> run the trained model on the test set.

  b) In multiple scenes experiment. The paper only showed the experiments on compressing two scenes, it's not clear how will the performance be and conclusion generalize to more scenes, e.g. train only one model on all 8 synthetic scenes together.

3. The paper also didn't consider the efficiency problem, training a Nerf model takes a long time, and also the author did not report the running time when doing inference on a trained nerf model, and do not have a comparison with HEVC+LLFF baseline. In literature, Nerf model takes more than one day to converge, while HEVC is very fast for both encoding and decoding.

Overall, considering the novelty and the experiment section in this paper, I don't think they are enough to reach the bar of ICLR, so I vote for rejection.

---

> ### Author Response · Authors · 2020-11-24
> **Response**
>
> Thank you for the points raised. Find our responses below:
>
> *A more valid baseline should be: receiver receives the training images compressed by HEVC -> decode the images -> receiver train a new Nerf on the decoded images -> run the trained model on the test set.*
>
> Although this is a valid scheme, we do not consider this as a practical baseline, since it requires the receiver to incur a prohibitively large cost in training a new NeRF model. The decoding time for the receiver is of primary concern, due to the fact that one compressed scene can be decompressed and used by an unlimited number of receivers. As such, a long encode time is potentially acceptable, whereas a long decode time is less so. So we only consider methods that fit these restrictions, and the compressed NeRF model and the HEVC + LLFF baseline both have decode times that are an order of magnitude or more faster than training a NeRF model.
>
> *...the author did not report the running time when doing inference on a trained nerf model, and do not have a comparison with HEVC+LLFF baseline*
>
> The inference times are roughly comparable between the two methods, in that they both require a forward pass through neural networks of comparable size. There has been no substantial optimization done for inference time in either method, so we do not provide detailed runtimes.
>
> *Nerf model takes more than one day to converge, while HEVC is very fast for both encoding and decoding.*
>
> This is true, although the LLFF model still needs to be trained. The advantage of the HEVC + LLFF method is that it can be used on unseen scenes, whereas NeRF has to be trained for each individual scene. Although (as we have said above) since this is a price paid by the sender, we do not think that this makes the method impractical.

---

> > ### Comment · AnonReviewer2 · 2020-11-25
> > **Thanks**
> >
> > Thanks for the rebuttal, however, I still have concerns
> >
> > 1. I do agree the baseline I proposed will be time-consuming in the decoder side, however, this is the only comparison I can think of that can demonstrate the effectiveness of the proposed method (for 3D scene compression), The author argues the training of nerf would take time in the sender side, but this can be mitigated by first training the nerf (once), then can run any (infinite) numbers of inference.
> >
> > 2. Can the author provide the exact numbers to convince us the time are comparable?
> >
> > 3. What about the issues I raised in the multiple scene experiments?

---

### Official Review · AnonReviewer1 · 2020-10-29
**Compressing NeRF's for single or multiple scenes**

**Rating:** 4
**Confidence:** 3

**Review:**

This paper proposes a method of compressing neural radience fields (NeRF's) by learning mappings from latent codes to model parameters such that both distortion/reconstruction quality and the rate get minimized. While maintaining the same level of quality, this method is able to compress NeRF models for more efficient sender-to-receiver transmission.

The strong aspects of this paper include:
(1) it addresses the valid problem of compressing NeRF models, which is particularly valid given that one usually trains one NeRF per scene. Without compression, the storage of NeRF's will grow dramatically as more scenes are considered;
(2) it achieves the same level of rendering quality, while compressing the model by 8-9x;
(3) it might have implications to future work that attempts to render NeRF's in real time or under a sender-receiver setup.

In terms of drawbacks, this paper can be made stronger by addressing the following points:
(1) while model sizes are compressed significantly at almost no cost of rendering quality, there is no improvement in rendering speed. In fact, because of the additional decompression, rendering novel views might very well take longer than the original NeRF. Since improving the NeRF rendering speed is vital to eventually achieving real-time rendering, I consider this a major drawback of this work;
(2) another major drawback in my opinion is that the work does not demonstrate its usefulness under the multi-scene setup. As the original NeRF is per-scene, and people desire a multi-scene variant of NeRF, the work would be much more impactful if it provides significant compression while achieving the same quality in such setups. Unfortunately, Fig. 5 shows having separate NeRF's have higher rendering quality at comparable sizes as the compressed models;
(3) it would be interesting to explain how this is related to meta-learning. One can imagine having a meta-NeRF that quickly adapts to different scenes. This also achieves similar compression effects, and may work better for the multi-scene setup. Maybe comparisons can be made to such meta-learning methods; and
(4) the work heavily relies on prior works in the compression aspect, and it's unclear to me what the novelty is in that regard.

---

> ### Author Response · Authors · 2020-11-24
> **Response**
>
> Thank you for the points raised. Find our responses below:
>
> *... there is no improvement in rendering speed. In fact, because of the additional decompression, rendering novel views might very well take longer than the original NeRF...Since improving the NeRF rendering speed is vital to eventually achieving real-time rendering, I consider this a major drawback of this work*
>
> The goal of our paper is not to improve the rendering speed of NeRF. Our method of compressing a representation function could be applied to other scene rendering techniques, many of which would have faster rendering times. Indeed, there have been some recent works such as Neural Sparse Voxel Fields (NSVF) which claim to improve the rendering speed of NeRF up to 10x. Our method would apply equally to NSVF as to NeRF. As such, we don’t think that our work should be judged in a negative light because of the limitations of the NeRF method (such as rendering speed).
>
> Training the compressed NeRF model does take longer than training the uncompressed model, since we have the extra computation in the calculation of gradients for the differential entropy term. However, after training, using the compressed NeRF model requires only a very small amount of extra computation as compared to the uncompressed model. This is because we don’t have to calculate the entropy penalty, and just have to use the pre-computed discrete probability tables to decode the weights - the time this takes is a negligible fraction of the current rendering speed.
>
> *the work does not demonstrate its usefulness under the multi-scene setup*
>
> Whilst it is true that the separate NeRF models perform slightly better at high bitrates, the multi-scene compressed NeRF models do achieve a significant improvement at low bitrates. Given that we are free to use separate or multi-scene compressed NeRF models for different bitrates, the multi-scene model therefore strictly improves the rate-distortion performance of the overall method.

---

### Author Response · Authors · 2020-11-24
**A message to all the reviewers**

Many of the reviewers raised concerns about the novelty of our work. Rather than repeating the same points in individual responses, we address that point here.

Although we have not introduced many new architectures or designs in our work, we still believe there is novelty in our method. We are taking a very non-standard approach to compression of visual media - we compress the 3d scene not via compressing views or other data directly, but instead by compressing an implicit representation function, from which the views can be reconstructed. To our knowledge we are the first to seriously study this kind of compression pipeline.

The methods we use (from model compression and neural rendering) are well established. But the combination of these and the problem setting (of minimizing the message length used to describe a full 3d scene) are not obvious to study. We are the first to do so, and we do achieve good experimental results. Our results are interesting even from a model compression perspective, since model compression techniques are almost always applied to classification problems, for example CIFAR and ImageNet. The problem setting for our model is a difficult regression problem, mapping dense points in space to a radiance field, which is characteristically different to (and probably harder than) the classification settings usually seen. The fact that the model compression works so well on this rendering function is significant, since it has not been shown before that such models can be effectively compressed.

It is also worth mentioning that our method of compressing a representation function would apply equally well to other media types, e.g. 2d images or audio. As such, we believe there is value in the community being made aware of this general methodology.

---

### Decision · Program_Chairs · 2021-01-07
**Final Decision**

**Decision:**

Reject

**Comment:**

Description:
The paper presents a method for encoding a compressed version of an implicit 3D scene, from given images from arbitrary view points. This is achieved via a function, learning with a NeRF model, that maps spatial coordinates to a radiance vector field and is optimized for high compressibility and low reconstruction error. Results shows better compression, higher reconstruction quality and lower bitrates compared to other STOA.

Strengths:
- Method for significantly compressing NerF models, which is very useful since such models are often trained for every new scene
- Retain reconstruction quality after compression by an order of magnitude

Weaknesses:
- The need for decompressing the model before rendering can be done means reduced rendering speed. This also requires longer training times.
- Experiments against other scene compression + neural rendering technique will have further strengthened the papers’s claims
- The techniques used are well established, and thus there is not as much technical novelty.